# The implementation, use and sustainability of a clinical decision support system for medication optimisation in primary care: A qualitative evaluation

Mark Jeffries[1,2]*, Nde-Eshimuni Salema[2,3], Libby Laing[3], Azwa Shamsuddin[4], Aziz Sheikh[4], Anthony Avery[2,3], Antony Chuter[5], Justin Waring[6], Richard N. Keers[1]

1 Centre for Pharmacoepidemiology and Drug Safety, Division of Pharmacy and Optometry, School of Health Sciences, University of Manchester, Manchester, United Kingdom, 2 NIHR Greater Manchester Primary Care Patient Safety Translational Research Centre, Manchester Academic Health Sciences Centre (MAHSC), University of Manchester, Manchester, United Kingdom, 3 Division of Primary Care, University of Nottingham, Nottingham, United Kingdom, 4 Usher Institute, Edinburgh Medical School, University of Edinburgh, Edinburgh, United Kingdom, 5 Haywards Heath, West Sussex, United Kingdom, 6 School of Social Policy, Health Services Management Centre, University of Birmingham, Birmingham, United Kingdom

* mark.jefferies@manchester.ac.uk

**Data Availability Statement:** All relevant data are within the manuscript and its Supporting information files.

## Abstract

### Background

The quality and safety of prescribing in general practice is important, Clinical decision support (CDS) systems can be used which present alerts to health professionals when prescribing in order to identify patients at risk of potentially hazardous prescribing. It is known that such computerised alerts may improve the safety of prescribing in hospitals but their implementation and sustainable use in general practice is less well understood. We aimed to understand the factors that influenced the successful implementation and sustained use in primary care of a CDS system.

### Methods

Participants were purposively recruited from Clinical Commissioning Groups (CCGs) and general practices in the North West and East Midlands regions of England and from the CDS developers. We conducted face-to-face and telephone-based semi-structured qualitative interviews with staff stakeholders. A selection of participants was interviewed longitudinally to explore the further sustainability 1–2 years after implementation of the CDS system. The analysis, informed by Normalisation Process Theory (NPT), was thematic, iterative and conducted alongside data collection.

### Results

Thirty-nine interviews were conducted either individually or in groups, with 33 stakeholders, including 11 follow-up interviews. Eight themes were interpreted in alignment with the four NPT constructs: Coherence (The purpose of the CDS: Enhancing medication safety and

**Funding:** This study was part of the NIHR
PROTECT Programme grant. PRoTeCT is funded
by the NIHR Programme Grants for Applied
Research Programme (1214-20012). The views
expressed are those of the author(s) and not
necessarily those of the NIHR or the Department of
Health and Social Care.

**Competing interests:** The authors have declared
no competing interests exist.

improving cost effectiveness; Relationship of users to the technology; Engagement and communication between different stakeholders); Cognitive Participation (Management of the profile of alerts); Collective Action (Prescribing in general practice, patient and population characteristics and engagement with patients; Knowledge);and Reflexive Monitoring (Sustaining the use of the CDS through maintenance and customisation; Learning and behaviour change. Participants saw that the CDS could have a role in enhancing medication safety and in the quality of care. Engagement through communication and support for local primary care providers and management leaders was considered important for successful implementation. Management of prescribing alert profiles for general practices was a dynamic process evolving over time. At regional management levels, work was required to adapt, and modify the system to optimise its use in practice and fulfil local priorities. Contextual factors, including patient and population characteristics, could impact upon the decision-making processes of prescribers influencing the response to alerts. The CDS could operate as a knowledge base allowing prescribers access to evidence-based information that they otherwise would not have.

## Conclusions

This qualitative evaluation utilised NPT to understand the implementation, use and sustainability of a widely deployed CDS system offering prescribing alerts in general practice. The system was understood as having a role in medication safety in providing relevant patient specific information to prescribers in a timely manner. Engagement between stakeholders was considered important for the intervention in ensuring prescribers continued to utilise its functionality. Sustained implementation might be enhanced by careful profile management of the suite of alerts in the system. Our findings suggest that the use and sustainability of the CDS was related to prescribers' perceptions of the relevance of alerts. Shared understanding of the purpose of the CDS between CCGS and general practices particularly in balancing cost saving and safety messages could be beneficial.

## Background

Medication safety is of primary concern to healthcare provision worldwide as described by the World Health Organization's Third Global Patient Safety Challenge: Medication without harm (2017) [1]. In the UK, there are more than a billion medicines prescribed annually in the community, hence prescribing errors in general practice are an important and expensive cause of preventable safety incidents, illness, hospitalisations and deaths [2]. It is therefore important to enhance the safety, quality and cost-effectiveness of prescribing. One approach to detect potential errors involves using 'prescribing safety indicators,' which are statements of potentially hazardous prescribing events that may place patients at risk of harm. These indicators can be deployed in general practice electronic health records to routinely identify patients at risk of hazardous prescribing (e.g. probable prescribing errors, whereby risks are likely to strongly outweigh any potential benefits) [3]. Prescribing safety indicators can be incorporated into information technology (IT) based tools which can then be utilised to support general practitioners (GPs) and other health professionals to improve the quality and safety of prescribing [4, 5]. However, the use of IT has not always been successful in the UK National

Health Service (NHS) with technology often failing to integrate or has been perceived as not adding value [6]. Such success or failure in implementation has in the past been attributed to the complex interactions between organisational processes, the technology and users [7]. Engagement between the health professionals working in clinical settings and the technology may also inform understanding of the potential impact on changes in working practices, roles and responsibilities [4, 8].

The Clinical Decision Support (CDS) system evaluated here was developed commercially and has been in use in general practices across England and Wales since 2014. The CDS provided a suite of messages that appeared to prescribers as alerts at the point of prescribing. This suite of messages is managed by local Clinical Commissioning Group (CCG) medicines management teams whereby individual or groups of messages may be switched on or off so the messages seen in practice could be tailored to local needs, guidelines or policies. CDS systems are commonly used to provide support for decision making by clinicians in both secondary and primary care settings. These tools may provide patient related information and clinical knowledge in order to enhance patient care, improve the quality and cost-effectiveness of prescribing and may reduce medication error or the potential prescribing of inappropriate medications [9–11]. CDS systems can be utilised to provide decision support alerts when a prescribing decision is being made with the aim of preventing a hazardous prescription. Available evidence indicates that this approach is moderately successful, but prescribers may override these alerts due to high volumes ('alert fatigue') and particularly if they interrupt clinical workflow by arriving at inappropriate times in the consultation [12–15].

CDS interventions may have different active interacting components that may act synergistically to yield one or more mechanisms of change and hence outcomes in changeable systems and are thus considered to be 'complex' [16]. Process evaluations can be conducted to understand more clearly why complex interventions produce observed effects, why these outcomes may differ from what was expected, what influence context has on these outcomes and what lessons can be learned for wider implementation and optimal use [17–19] It is also suggested that the use of implementation science and theoretical frameworks is important to support the robustness of process evaluation [17]. One such framework is Normalisation Process Theory (NPT) which focuses upon the work people do to integrate interventions into practice [20].

Previous studies designed to evaluate CDS systems have shown that the application of CDS might improve care outcomes, improve the process of care or have explored the acceptability of systems to professionals [10, 11, 20]. Much previous research has focused upon hospital settings [10, 11, 15]. With a notable exception in Trinkley et al [21], who explored clinician preferences for their ideal CDS that could be utilised in primary care, there has been little in the way of qualitative research that has explored the complex and interrelated sociotechnical processes involved in the implementation of a CDS intervention nor examined their longer term sustainability. This study therefore aimed to understand the factors that influenced the successful implementation and sustained use in English primary care of a CDS designed to enhance the quality and safety of prescribing.

## Methods

### Study design

This study was a qualitative process evaluation using semi-structured interviews of multiple stakeholders working in primary care settings in England. The study took place across general practices within four separate CCGs, or partnerships of CCGs, in the North West and East Midlands regions of England which were chosen to provide different geographical characteristics. CCGs are the clinically led, authorised NHS bodies that plan and commission healthcare

services for their local area [22]. CCGs had pharmacist-led medicines management teams that supported quality prescribing and medication safety across their area.

## Sampling and recruitment of participants

The sampling frame was stakeholders involved in the implementation of the CDS system, and/or in the ongoing use of the system, working within the CCG areas. General practices and CCG areas were only included if they had installed the CDS within the previous twelve months. In addition, staff working for the CDS developers were also approached to take part in interviews. Stakeholders therefore included CCG managers, CCG pharmacists and pharmacy technicians, GPs, general practice nurses, general practice-based pharmacists and pharmacy technicians and CDS developer staff. CCG and CDS developer staff were approached directly by MJ or RNK via email or telephone and invited to take part in interviews. General practices where staff could be invited to take part were identified through discussions with CCG managers. A typology framework of practices in each CCG was developed based upon demographic factors to facilitate recruitment of diverse groups–including size of practice, use of different electronic health record (EHR) systems, indices of multiple deprivation for the area the practice served, CCG reported engagement with the CDS, and time since implementation of the CDS. Potential practices were approached by email or telephone by MJ and invited to take part. In addition, MJ visited CCG meetings of groups of general practices. Once practices indicated that staff might be interested in taking part, MJ visited the practice to explain the study further and provided written information. Practices consented to take part in the study before individual staff were approached. Individual general practice staff were approached directly by telephone or email, or through liaising with the general practice manager. A total of 41 practices were approached, 32 either declined to take part or did not respond to approaches and nine practices consented to take part. Reasons for not taking part were usually associated with time and staffing commitments. All individual potential participants were provided with study information and given at least 24 hours to decide if they wished to take part. They were then contacted by MJ to arrange a convenient time for interview.

## Data collection

The semi-structured interview schedule was informed by NPT [20] and developed through discussions between the authors and reading of relevant literature [4–8, 14]. The interview schedule (included as S1 Appendix–Interview Topic Guide) was designed to illicit how people understood the value of the CDS, how it had been implemented, the work in using the system and how the intervention might be sustained. The interview schedule was reviewed iteratively throughout data collection to ensure it continued to be appropriate. Semi-structured interviews were conducted with a range of stakeholders–CCG staff including Pharmacists and Pharmacy Technicians, GP Staff including GPs, Nurses and Pharmacists and staff from the CDS developers. Follow-up interviews were conducted approximately 12 months after the first interview with 11 participants who had taken part in first interviews. These participants were purposively selected to provide a range of different stakeholders, in order to understand changes that may have occurred and how the intervention was being sustained in everyday practice beyond the initial intervention period. All interviews were conducted solely by MJ, a researcher with extensive experience in conducting research related to medication safety and the use of technology and expertise in qualitative research interviewing, Interviews took place at the participants' usual place of work (general practice, CCG offices) or at university premises (two interviews). No interviewees were known to MJ prior to the interview. Thirty interviews were conducted face-to-face and 10 by telephone. As approved by the ethics committee

participants were offered a £20 shopping voucher per interview as reimbursement for their time. Vouchers were accepted by all staff in general practices, who took time out from their working day to participate, but declined by eight of the ten CCG staff.

## Theoretical framework: Normalization process theory

Theoretical frameworks for technological interventions have seen technology, people and organisations as separate things which operate at multiple levels but do not do so through interaction and have often regarded these as separate homogenous elements [7]. Technological intervention can be observed in ways that understand it as part of complex and interdependent social processes [6, 7]. Normalization Process Theory (NPT) has been used to understand the ways new technologies and work practices are implemented and focuses upon the social processes of implementation [4, 20]. Interventions involve changes in the actions that people do so that the new behaviours involved in the new intervention become adopted [20]. NPT suggests that interventions are adopted and implemented because of the work and actions people undertake, both individually and collectively, to integrate the intervention into routine practice [20, 23–25]. NPT provides a rational framework and allows for an understanding of how individuals and groups make sense of a new intervention and work together to build confidence in the new practice and change their actions and behaviours in order to enable it to happen [20, 23–25]. Implementation of a new intervention involves changes in human interactions with material things, changes in human relationships, changes in the rules and social norms that make actions possible and changes in the ways systems are understood and defined [25].

NPT is built upon four constructs: coherence–the ways in which the intervention is understood, cognitive participation–the ways people work together to put the new intervention into practice, collective action–the ways people work to operationalise the new intervention, and reflexive monitoring–how people evaluate the new practice and work to sustain it. These constructs are further described in Table 1. NPT has been used in a number of evaluations of health care interventions including those for prescribing safety and patient digital feedback

**Table 1. Normalisation process theory constructs [22–25].** From Jeffries et al PLOS ONE 2018.

| NPT constructs | |
|---|---|
| **Coherence** | **Cognitive participation** |
| The ways in which people define the intervention and how they make sense of it. | How people organise themselves and others through relational work to implement the new intervention. |
| The work people do to understand how the intervention might involve a different set of practices. | Working together to collectively contribute to the new ways of working that the intervention requires. |
| The work people do together to understand the intervention and to integrate it into a healthcare setting. | How the actions and procedures that will be needed to sustain the intervention are defined. |
| The individual tasks involved in the intervention and what people do to attribute worth to that new practice. | |
| **Collective action** | **Reflexive monitoring** |
| How the intervention is operationalised and enacted in practice. | How other individuals and groups evaluate the intervention and look to sustain it. |
| The collective and interactional work that people do with each other in order to adopt the new intervention into practice. | The work that participants do individually and collectively to evaluate and determine how effective the intervention is for them and for others. |
| The knowledge work to build confidence in the new practice, divisions of labour and allocation of work including the tasks people do and how those tasks are related to their existing skill sets. | The impact of the new practices upon their own work. This may include attempts to modify the intervention. |

and has been said to be particularly useful for drawing out the multi-faceted nature of interventions [4, 26, 27]. In recent research NPT was considered useful in providing insights into how relationships and communication between health professionals could help drive the implementation and sustainability of a technological intervention utilising an audit and feedback system for medication safety in primary care [4]. It was therefore considered an appropriate theoretical framework through which to explore the use of a CDS in primary care.

## Data analysis

Data analysis was informed by NPT as a theoretical framework. MJ led the data analysis, which was thematic and conducted alongside data collection. The analysis followed two parts: firstly, an inductive thematic approach informed by Braun and Clark [28]. For the initial inductive thematic approach, MJ used QSR NVIVO 12 to organise the data and inductively coded six early interviews to identify features, patterns, groups of codes and potential themes to build a preliminary framework for application to the data and to inform further data collection. These six coded transcripts were independently read by RNK and further discussed by MJ, RNK, and JW. Next, analysis moved to a template approach [29] which involved developing a coding template as a consequence of these discussions and applied to 27 interviews. Further discussions of the transcripts, codes and themes were undertaken by RNK, MJ, NS, LL and AC and the coding template reworked as a result. Further development of the coding framework into themes was undertaken with all authors, consensus reached, and the codes and themes applied to the full data set. The themes were then mapped, integrated and interpreted alongside the four NPT constructs [25].

## Ethical approval and consent to participate

Ethical approval for the study was granted 10th October 2017 by the University of Manchester Research Ethics Committee. Further approval to conduct the research was granted by the NHS Health Research Authority (IRAS project ID: 233079) on 26th October 2017 and by the research and development offices for the NHS participating regions. General practices gave written informed consent for their staff to be approached to take part in the study. All interview participants at general practices, CCG staff and the CDS developer staff gave written informed consent to take part in the study, and for interviews to be audio-recorded and transcribed verbatim.

## Results

Thirty-nine interviews were conducted with 33 participants. These comprised of interviews with general practice staff (n = 20:- GPs = 14, Nurse prescribers = 3, practice pharmacists = 3), CCG staff (n = 10: CCG Pharmacists = 7, Pharmacy Technicians = 2, data analyst = 1) and CDS developer staff (n = 3). Of the 39 interviews 11 were follow-ups. Thirty-four interviews were conducted one-to-one and 5 as group interviews. The 5 group interviews each involved staff working at the same CCG or general practice (Group interview 1—CCG pharmacist and CCG Pharmacy Technician; Group interview 2–3 GPs; Group interview 3—GP and GP Pharmacist; Group interview 4—CCG Pharmacist and CCG data analyst; Group interview 5—follow up to group interview 3 with GP and GP Pharmacist). Stakeholder interviews are further described in Table 2. The first interviews were conducted between December 2017 and November 2019 with follow-up interviews conducted between December 2018 and June 2020. Interviews ranged in duration from 13–68 minutes with a mean length of 36 minutes. The main findings are mapped onto the four NPT constructs as described in Table 3.

**Table 2. Participants by role and region.**

| PARTICIPANTS—By Role and CCG region | Region 1 | Region 2 | Region 3 | Region 4 | No region | TOTALS PER ROLE | |
|---|---|---|---|---|---|---|---|
| **PRESCRIBERS** | | | | | | | |
| GP | 2 | 1 | 3 | 8 | 0 | 14 | GP |
| Nurse Prescriber | 1 | 0 | 2 | 0 | 0 | 3 | Nurse Prescriber |
| Practice Pharmacist | 0 | 0 | 1 | 2 | 0 | 3 | Practice Pharmacist |
| **TOTAL PRESCRIBERS** | **3** | **1** | **6** | **10** | **0** | **20** | **TOTAL PRESCRIBERS** |
| **STRATEGIC OVERVIEW–CCG OR CDSDEVELOPERS** | | | | | | | |
| CCG, Manager Pharmacist or Pharmacy Technician | 2 | 2 | 3 | 3 | 0 | 10 | CCG, Manager Pharmacist or Pharmacy Technician |
| CDSDevelopers | 0 | 0 | 0 | 0 | 3 | 3 | CDSDevelopers |
| **TOTALS STRATEGIC OVERVIEW** | 2 | 2 | 3 | 3 | 3 | **13** | **TOTALS STRATEGIC OVERVIEW** |
| **TOTALS PARTICIPANTS PER REGION** | **5** | **3** | **9** | **13** | **3** | **33** | **TOTALS** |

## NPT construct: Coherence

The CDS system was defined and made sense of as a system that had the potential to enhance medication safety and improve cost effectiveness. The relationship between the users and technology contributed to prescribers understanding of the relative advantages and disadvantages of the intervention. The latter included perceptions that the CDS would slow the practice electronic health record (EHR) or otherwise impact upon workflows. Engagement and communication were part of the collective work that people did to understand and integrate the intervention into primary care.

### The purpose of the CDS: Enhancing medication safety and improving cost effectiveness

Participants saw that the CDS could have a supportive role in enhancing medication safety and it was described as not the '*ultimate solution for helping with prescribing, but it complements (it)*' *(CCG Pharmacy technician 2)*. It was perceived by the CDS developers that

**Table 3. Findings mapped to NPT constructs.**

| NPT constructs | |
|---|---|
| **Coherence** | **Cognitive participation** |
| Participants saw the CDS systems as enhancing medication safety and improving cost effectiveness. | People worked together to manage the profile of alerts to ensure timely and relevant alerts would be received in general practices and to avoid alert fatigue. |
| The ways people worked to integrate the CDS system into existing technology and workflows. | |
| Engagement and communication between different stakeholders was seen as important to integrate the intervention into practice. | |
| **Collective action** | **Reflexive monitoring** |
| Within general practices staff worked to operationalise the intervention. This involved making prescribing decisions in using the CDS that took into account patient and population characteristics. | At CCG level work was required to adapt, change and modify the CDS system to ensure it continued to be used optimally. |
| Using the CDS involved work to balance the new knowledge found in the intervention with prescribers own knowledge, experience and judgement. | The use of the CDS involved learning and behaviour change. |

prescribers had confidence that the alerts in the CDS were providing a safety net, and that '*(The CDS) it's like having a medicines management pharmacist or technician sitting by your side, looking at your choice of medicine.*' *(CDS developer pharmacist 1)*. There was trust in the system that it could provide accurate messages and had the capacity to provide the information needed at the point of prescribing. The triggering of alerts had the potential therefore to improve care, and participants highlighted this as important reasons to implement and use the system.

> *I'd hope that it would be making things better for them because the safety net is there. [. . .]I feel a little bit more confident that there are those safety messages there and not relying solely on the clinical systems where sometimes there isn't an alert.*

> *(CCG Pharmacist 2)*

The value of the system was considered in terms of the quality of care that was potentially going to be realised, balanced against the cost of the system. Value was seen in terms of safety since the '*real positive affect of it is probably the safety*' *(GP13)*. Cost was expressed as financial and in workload or time, so cost savings at the point of prescribing might free CCG medicines management teams to undertake other quality and safety work, and that going into practices to '*do an audit and change it (prescribing practice) (is) quite time consuming*' *(CCG Pharmacist 5—Follow up)*. Financial considerations were considered important to CCGs, and as business obligations to general practices. Cost was perceived as a driver for the CCGs and part of necessary financial considerations regarding cost savings and budgets. However, CCG medicine management teams talked about wanting to implement a tool that went beyond '*just making financial recommendations*' *(CCG Pharmacy technician 1)* to something that could incorporate safety, best practice and quality. Without the cost saving element of the system it was felt that there would not be an incentive for the CCGs to invest in the system and sustain the intervention in future.

> *And a cost-saving tool. I think if there wasn't a cost-saving attached to it, the CCG wouldn't fund it. I think if it just sold itself on safety, then I don't think the CCGs would invest in it. I know that sounds really bad but the way that the finances are at the moment, they can't afford to be investing in things that aren't going to show a return on investment. So, it's just fortunate that it can show a return on investment so that then we can invest in the safety side of it..*

> *(CCG Pharmacist 5)*

Whilst health professionals, and particularly prescribers in general practice, recognised the need for cost savings they preferred a focus on safety.

> *[. . .] for me personally (the main focus has) got to be patient safety. Because if (the CDS alerts) can stop you prescribing something that would do the patient harm, that otherwise wouldn't have flashed up to you, that's got to be worth its weight in gold from my perspective. . .*

> *(GP14)*

Similarly this prescriber perceived that cost-savings gained through making prescription switches represented short-term thinking. Whilst acknowledging that the cost saving was necessary, since public funds were involved, it was thought that simple switches of medications did not fit with a long-term strategy.

*I don't think cost is irrelevant. The reason I get frustrated about cost is because it changes so much. And there's a real short-termism about budgets. [. . .]. . . we have to look for cost savings and be aware that we're spending tax payers' money and that we can't just ignore that, but on the other hand it doesn't always feel like there's a consistent long-term strategy that we're moving people in an organised way towards the more cost-effective prescribing strategy and that it will stay like that.*

*(GP4)*

## Relationship of users to the technology

The relationship between users and the technology was important particularly in the way the CDS was integrated into both existing technology and usual workflows. It was considered important that the CDS fitted well with the general practice clinical system, with some participants reporting that it was slowing the clinical system down or distracting the prescriber from the patient. Changes to the practice EHR were reported to impact on prescribing and associated tasks since it was '. . . *a lot more complicated.' (GP12_ Follow-up)*

This GP reflected that it was important for their practice that the CDS system didn't impact upon workflows.

*We got told, that [. . .]it was similar to what we already had anyway, and so we shouldn't, besides the screens looking a little bit different, it shouldn't really impact on our workflows or anything like that. And essentially, by and large, they were right. It had a few little extra bangs and whistles, particularly I think, to connecting to links to websites, if we wanted to know more, which seemed a bit more better integrated, so we had all the functionality of previously, plus a bit more, and it was a bit tidier. . ..*

*(GP5)*

GPs talked of the CDS being initially '*quite clunky and quite intrusive' (GP10—follow-up)* to use. For GPs, the continued use of the CDS was supported by it being further embedded and less obtrusive to the user. After this early period, between 6–12 months after implementation, the CDS was seen as easy to use, unobtrusive, just 'part of the system' and that it was easy to navigate through it.

*I suppose it's just become more embedded so I am a little bit less aware of it, I suppose, it's become a normal part of the clinical system that I use. And I'm definitely aware of it still but I don't think, the comments I would have now haven't probably evolved a great deal,.*

*(GP4-Follow-up)*

## Engagement and communication between different stakeholders

Early engagement with GPs and CCGs by the CDS developer was seen as positive in helping prescribers see the usefulness of the system and therefore encourage implementation.

*So we try and get the GPs engaged early on because what we found is, if an organisation just buys it and chucks it out, GPs are like, hang on a minute, you know, yet again you've bought*

*something that's affecting us and we don't have any buy-in. Whereas if we go and we push the fact that it's best practice and it won't pop up a lot, and if it pops up, there's normally a reason . . .*

**(CDSdeveloper pharmacist 1)**

Engagement between the CCG and general practices was considered important to the implementation particularly to *'make sure practices felt involved' (CCG Pharmacist 3)*. It was felt that it was important for CCG staff to communicate with practices through their practice manager and that this would be an effective strategy *"Because they (practice managers) were key to getting their practices on board" (CCG Pharmacist 3)*.

At CCG level, the successful implementation of the CDS was reported to be dependent on providing support for general practices. Without such support it was anticipated there would be disengagement.

*You couldn't just put it into a practice with no support at all. They'd be like, well, if I'm getting a message and I don't understand it, I need somebody who's a representative to say, I don't get that, is there something I'm missing or is that message wrong? So, I think they need a personal touch and especially right at the beginning when it gets turned on. I think they need to know that they've got back-up because if it all goes belly up and they're getting messages left, right and centre, they need somebody there to support them. Otherwise, I think they'd just turn it off.*

**(CCG Pharmacist 5)**

There was a divergence in viewpoints here in how the CCG staff and GP staff saw the communication between them. Despite CCG staff talking of the support they provided to GPs, general practice staff throughout the interviews commented that it was not unusual to see 'top down' implementation of the CDS with minimal induction or training.

*No, just, these things happen, that's not out of the ordinary [. . .]. So, yeah things just happen [. . .] They often will appear before you have any training or knowledge of it whatsoever. Then if you're lucky maybe there's a launch event six months after you've already started using it.*

**(GP14)**

Whilst prescribers reported that training was often minimal and that the system may have only had brief introductory explanations from the CCG implementing it, some commented that training may not have been required as the CDS was not complicated to use.

*I think [..] somebody from the CCG might have come to one our neighbourhood meetings and told us it was happening. But I don't remember any kind of training and they said, it would be pretty much like it used to. . .There may genuinely have been a five minute. . .[. . .] And I think somebody from medicines management may have briefly said, it will be rolled out, it's pretty self-explanatory, when this pops up it's pretty much like, and that was it. But then it's not a complicated. . .why would you.*

**(GP6)**

Minimal training and 'top down' intervention was reported to lead to some disengagement between the CCG and GP practices. In a follow-up interview exploring the continued and

sustained implementation of the system, one CCG medicines management pharmacist suggested that the CCG *'could probably improve GP involvement' **(CCG Pharmacist 5)**.* Elsewhere, this GP felt there was little in the way of direct conversation with the CCG but that any engagement most likely would be through the general practice-based pharmacist.

*I'm not aware of anything. [. . .] I'm not aware of any direct contact. We do now have a pharmacist [. . .] she's been aware of being able to access things that we've, you know, the log essentially, so the things that we have overridden or the list of things that we have actually changed as a result of (the CDS system) messages. So, she must have had some level of communication with the CCG at that level to be aware that she can interrogate our data in that way.*

**(GP4_Follow-up)**

Communication with the CDS developers from the CCGs was perceived to be important in getting changes made to the CDS alerts over time but could add additional workload for the CCG. As this CCG pharmacist explained, getting changes to the formulary and getting the CDS messages ratified and agreed upon took time.

*So in terms of the account manager I think our relationship with and access to (name) is fine. She does seem to be able to get things moving when needed. So with the formulary we have to wait for that decision to go through the area prescribing committee, then we have to wait for it to be ratified, then we have to wait for the minutes to go online, and only then can we amend the formulary. So that process can be anything up to six weeks. So there's always that period of time when something's actually on formulary but nobody knows about it.[. . .].*

**(CCG Pharmacist 4)**

## NPT construct: Cognitive participation

By careful management of the profile of alerts, people worked together and contributed to new ways of working to implement and sustain the intervention. This was undertaken through the management of the profile of alerts centrally from the CCGs.

### Management of the profile of alerts

Profile management was considered essential to running the system but was described as resource intensive in terms of '*the amount of work you've got to put into it is quite immense*' **(CCG Pharmacist 5)** and that it could not be done '*without a team of people working on it*' **(CCG Pharmacist 5)**. It was felt that without active profile management, GP staff using the system would become '*so fed up with it, they'd just turn it off*' **(CCG Pharmacist 5)** because it would provide too many alerts which as this GP reflected would lead to them ignoring them.

*So if there is a something that pops up, I suppose there is a higher chance of me looking it up but I'm just aware of the fact that if there is loads of red mess, almost my default mechanism is to go, oh pile of shit, you know and ignore it which actually is probably the very worst thing I could be doing. I'm really not sure that. . .I know I have never sat there and laboriously gone through all the alerts.*

**(GP6)**

It was also seen as important to avoid the duplication of alert messages within the CDS system. It was felt that it was important to maintain the continuity of messages that came from different systems–from either the CDS or the practice EHR. Such simplification of messages and the avoidance of duplication made management of the profile of alerts more straightforward and avoided extra resource input.

*. . .there's a huge amount of resource goes into that, there's a lot of resource that goes into (the CDS system), and actually I have this big thing about duplication and simplification at work and actually really don't like doing things twice. [. . .] (The CDS system developers) are quite strict on the fact that actually they don't author messages if it's in the host system they can't write new messages, which is right. And ultimately again it's in one place, you don't want it in another. So it's sort of maintaining the continuity as best we can really.*

**(CCG Pharmacist 4)**

One CCG pharmacist commented that having alerts already embedded in the CDS was less time consuming and that the messages had more reassurance in that they were national ones received by all. This made it easier to maintain the profile and provided reassurance that everyone was receiving the same message.

*In the profile maintenance of it, because we had to author our own messages before, whereas with this one, a lot of them came authored already, based on national guidance, so the actual maintenance of the profile was a lot easier. And I think that was the biggest driver for us, was that actually, yes, it's a great tool, but how much work would we need to put into it to maintain the profile? But because a lot of that work was done, and also, from a variation point of view, because the messages were from national guidance, you can feel reassured that actually everyone's getting the same message, as opposed to having locally authored messages, which might not be in sync with what other people are doing.*

**(CCG Pharmacist 6)**

There were different approaches and processes involved in the management of the alert profile. Some CCGs adopted a gradual approach to implementation without all the messages in the profile enabled, and then checked that these messages were useful for their population and the work they were trying to do. This approach from CCG medicines management teams was deemed useful in achieving good acceptance of alerts and could mitigate against alert fatigue, allow for the alerts to be tailored for the local population and for the system to complement other work that the CCG might be undertaking.

*Well, they [the CDS developers] gave us a choice. They said we could go with a profile that everything's on, but not enabled, so we would go through and switch on what we wanted. That was one choice. Or we could, more or less, have that same profile with it all enabled straightaway. So, we decided to go step by step and just enable what we wanted, because of the history we had with pop-up fatigue and we didn't want to overdo it. Plus, even the time capacity, just to go through together what we wanted enabling, initially, which took time, which we didn't have, so we went through very slowly, step by step. And it worked really well*

**(CCG Pharmacy technician 2)**

It was also reported that having messages linked to the individual patient was a way of avoiding alert fatigue.

*Plus, we had more guarantees that things were triggered to more of an individual status, whereas the original software [from a different developer] we had, the messages that we put on, whether be it cost, quality, whatever, it would flag all the time, so it wouldn't be individualised to a person. And we found a lot of GPs and clinicians would get pop-up fatigue; they'd get so fed up of it all.*

**(CCG Pharmacy technician 2)**

## NPT construct: Collective action

Within general practices staff worked to operationalise the intervention. This included the ways in which they responded to the alerts within the CDS according to the needs of patients and patient populations. Responding to the alerts was related to prescribers existing skill sets in that the knowledge base within the CDS was balanced against their own expertise.

### Prescribing in general practice, patient and population characteristics and engagement with patients

The response to alerts and prescribing decisions were reported to be influenced by patient and population characteristics. This could particularly involve complex patients with multi-morbidity and polypharmacy and therefore prescribing acute medicines or those that a patient had been taking for some time. Prescribing decisions were often, particularly for more complex patients, made more holistically. One community matron talked of '*looking at each patient holistically from their disease management into their prescribing' (**Nurse prescriber 1)**. Similarly it was considered important, in responding to alerts, that patient choice was taken into consideration, to be mindful of the patient and look at the wider picture of safety for individual patients.

*Yes, I think I'd like to think I do involve them. Yes, very much so, because with diabetic drugs there's so many at the moment. It used to be two tablets and then go onto insulin and now it's just, the market is huge. It's a long-term condition so you can imagine the amount of drugs that they want. . .So yes, I do discuss with them and obviously, you know, for some people, they just won't want to put on weight, fair enough. So yes, I'm quite happy to prescribe what's right for the patient.*

**(Nurse prescriber 2)**

There was variation in perceptions as to whether the system benefited prescribing for these patient groups or not. Some GPs said that they were more likely to override the system if the patient had been taking '*something for a long period of time' (**GP10 -Follow -up)**. One GP felt that *'a more complicated patient is where it's actually in a way more useful'***(GP1)** since it was perceived that the system would help with the potential interactions between multiple medications. However other prescribers felt that polypharmacy meant that '*when you're doing the review it pops like all the time' (**GP2)** and '*just becomes overwhelming because obviously there are lots of issues that are potentially already going on with their other medications, so it all pops up at once'***(GP6)**. It was therefore felt that dealing with polypharmacy and multi-morbidity made the alerts from the system too numerous and required prescribers to become selective in which alerts they acted upon.

*I had over 300 patients on over 25 medications, and that is impossible for (the CDS system) to help me. I've got to be really quite selective about what I do. Because if I've got 20 alerts up when I'm doing a medication review I cannot deal with 20. But I've learned to be a bit more selective about it, and I'll go for the low hanging fruit, and I'll perhaps do a little bit in the notes to say discuss such and such next time.*

*(GP3)*

Such a distinction between prescribing acute medicines and making changes following a review was related to the perceived additional time and workload involved. This was particularly related to prescribing when completing repeat medications in situations when the patient would not be present.

*The difficulty comes when you're doing repeat medications and it pops up then and you just think, the faff of changing it midway through when a patient's already used to one thing, in order to do that you've got to contact the patient, you've got to explain to them, [. . .]we wouldn't do (that) as GPs because we just haven't got the time for it.*

*(GP12)*

## Knowledge

The CDS operated as a knowledge base allowing prescribers access to evidence-based information that they otherwise would not have. This was seen as important since prescribers were said to not be able to have full knowledge or could not be expected to know everything or "*look up every single drug that I'm going to prescribe*" *(GP9)*

However, there was a balance between the system and prescribers' '*own clinical judgement and experience as a doctor*' *(GP14)* and that prescribers would override the system if they felt on balance it was the better option for the patient.

*So to some extent I'm just using my judgment about that and then experience, so it's not always evidence-based, it may be they're just a personal prejudice about. . .prescribing habit comes in possibly and I think, well, yeah, fair enough, that's a risk that I take. That's a decision I make with a patient and I'm on this occasion not going to take notice of that because I think the indication outweighs the risk."*

[*GP4*]

Prescribers were thought of as having knowledge of patients and their medical history and an understanding of different medical interventions. Alerts were balanced against what the GP knew of the patient and that the patient had had no reaction in the past.

*If you prescribe something and there was an interaction between some other tablet, that's good. Yes. Then you see on probability if it is going to benefit patient or not. [. . .] Somebody who has asthma and or problems with their heart, they would benefit from the beta blocker and beta blocker can trigger it. So, it will come out in an alert. [. . .] But they already had before beta blocker and you're titrating, and the cardiologists and they didn't have any reaction, but the system doesn't know. The system is too persistent. It only does what it's been told to do. [. . .] So, the system doesn't know that. . .[. . .] and you need to explain to the patient. . .?*

[*GP2*]

It was said that general practice was about balancing risks and the response to alerts had to fit into that risk management which often involved a pragmatic approach.

*. . .. what we're trained to do as doctors, is to do the difficult stuff, being pragmatic, the balancing two, three, four, competing risks, and working out what to do despite them, and how to manage them, and how to safety net them and all that sort of stuff, and (the CDS system) definitely helps from that, because anything that helps remind you of what the risks are, helps you manage them better.*

*(GP13)*

Knowledge was gained through experience and it was thought that '*experienced GP[s], they know what they're prescribing and how they want to prescribe*' *(GP2)*. At least one GP felt that GP knowledge, training and expertise was more extensive than non-medical prescribers for whom the system might be more helpful. Others felt that GPs knew the obvious alerts but help with more subtle ones would be good. The alerts were perceived to be more relevant and more useful for more junior inexperienced colleagues and less useful for experienced GPs who had more expertise.

*Experienced GP(s), they know what they're prescribing and how they want to prescribe and the reaction of the medications and you're dealing with a certain number of tablets which you are familiar with. So, your prescribing habits changes very little. [. . .] However, with the younger generation, with inexperienced GP. . .you know, younger doctors, I think it's quite useful.*

*(GP2)*

## NPT construct: Reflexive monitoring

Maintenance and customisation of the system was important to how groups and individuals worked to sustain the intervention. The system was evaluated through the impact it had on learning and behaviour change.

### Sustaining the use of the CDS through maintenance and customisation

At CCG level, work was required to adapt, change and modify the CDS system to ensure it continued to be used optimally in practice and fulfilled local priorities. Profile management involved turning on and off different messages at different times of the year based upon seasonality in a 'rolling programme' over time to ensure both relevance and usefulness and to avoid prescribers becoming annoyed with the volume of messages.

*we're going to work a bit harder on making sure the messages [alerts] they do get are relevant to them and useful so we're going to be working harder on looking at the rejection reasons and tailoring the messages they get. We were also thinking about turning messages on and off so they don't get fed up with them so doing it on a rolling programme so that they align with the work that we're doing. For example, the self-care work so looking at over-the-counter products that patients should be using, we're doing that in waves. So, we started with dry skin, hay fever. Some of it's seasonal anyway so we can turn the messages on and off seasonally or when we're doing the work to remind them. So yes, just trying to use it a bit more savvily, if that's a word.*

*(CCG Pharmacist 5).*

Profile management was a dynamic process over time. New messages and alerts could be added to the CDS. This was seen as a solution to new issues raised locally in general practice, at secondary care or through connectivity between the CCG and general practices

*There might be things that come up locally. They might be mentioned at the hospital, that things that are coming through that shouldn't be, that aren't approved, and, oh right, we can maybe ask for that to be added onto (the CDS system). Or things at meetings; I've had two meetings today where I've got half a dozen things that could potentially go onto (the CDS system) to help our GPs.*

**(CCG Pharmacy technician 2)**

Where CCGs were aware that a message was commonly rejected, they would consider whether they needed it or not.

*. . .what we need to be looking at is what's firing and what's not being accepted and what's being rejected, and what percentage is being rejected and whether that message is still appropriate or not appropriate? Do we need to turn it off? So that's all the backend work that we still need to do. Because there's no point having a message on if 90 per cent of your users are [. . .] ignoring it. . .*

**(CCG Pharmacist 6)**

## Learning and behaviour change

The CDS had some potential for learning and could lead to behaviour change; the system was seen to provide education for GPs and other prescribers where alerts on prescribing safety indicators have been missed. Prescribers could dynamically adapt to the system and if they changed their prescriber alerts would no longer be seen, though this might involve second guessing the system.

*. . .and it's actually educational because a lot of them you know the warning before it comes up because you've seen it so many times so you're already aware of it. So I mean some of the things I know now are because of it, so there is an educational element to it [. . .] Yeah. Yes, definitely it does, yeah. And bringing up some of the risks as well, the long-term risks, fracture risks in certain drugs and things like that it's brought up, which you probably wouldn't have thought about at the time; so it's definitely beneficial from that point of view.*

**(GP1)**

For this practice pharmacist, the system had potential in helping prescribers to reflect upon their practice.

*It's active, it's there, it works as and when required. It's giving you information and just needs to be succinct. We also need to have a time where we can pull that information off on a monthly basis and reflect on the changes that we've made in order to make it safer [. . .] (The CDS system) has great potential to make things safer. We should be able to use it deeper, so we can reflect on our prescribing so that future prescribing is safer.*

**(Practice pharmacist 3)**

For some the potential for learning was seen as limited and might involve improved pre-scribing but no extensive change in working practices. Some participants commented that whilst learning might involve recognising the value and need for the prompts, the CDS did not involve enhancing understanding and learning but simply made changes.

*More times than not, I'd just follow it through. Just 'cause, it's the time factor. But, you know, on the odd occasion, when I've thought, oh, I wonder what that's about, that's possi-bly...'cause often it'll flag up something that's possibly a learning need for me. [. . .] Which, would have been nice, 'cause if I can then go to the page, I can then make a note of it, and then go away and read up about it [. . .] So, if you take the example of, I didn't know that beta blockers cause, or can make asthma worse. Oh, that's a doctor's educational need [. . .] I can go away and learn.*

*(GP5)*

## Discussion

This study revealed that participants valued and understood the CDS as having value and a role in medication safety and additionally as a. cost-saving tool. Engagement and communica-tion between general practice and CCG staff was understood as important for the intervention to ensure optimal use and drive successful implementation, adoption and sustainability. It became apparent that the majority of 'work' required to embed and then sustain the impact of the CDS was required at CCG level in supporting general practices to use the tool and adapting its alert profile over time. Participants felt this necessary to avoid alert fatigue and to ensure that appropriate, timely and relevant alerts were being displayed to prescribers that reflected the changing local landscape. This work at CCG level was considered a dynamic process that was demanding, time-consuming and required different approaches. In general practice, tak-ing clinical action in response to CDS alerts was influenced by the contextual background within which prescribing decisions took place. Actioning alerts was dependent upon an array of contextual factors that included the characteristics of patients, populations and prescribers, perceptions of time and workload and the engagement with and involvement of patients.

It has been previously reported, through qualitative research into CDS support alerts for medicines management, that clinicians prefer to have clinically relevant and pertinent infor-mation presented in alerts that do not interrupt their workflows [21] and participant views in our study were in concordance with this. Trinkley et al [21] similarly found that clinicians pre-ferred clinically relevant alerts that presented pertinent information and allowed for flexibility in response. Additionally, Van der Velde et al [30] found in a systematic review that CDS was more effective when patient specific suggestions were made as was the case with this CDS. Both Jia et al [10] and Monique et al [31] found that the specificity of alerts could improve alert fatigue. It is clear from our findings that CDS systems require ongoing maintenance to ensure that alerts are timely, relevant and reliable. The avoidance of alert fatigue was in part achieved by the continual monitoring, development and modification of the suite of alerts that individual CCGs actioned within their locality. Importantly, prescribers disliked those alerts considered irrelevant or trivial and these were perceived as overwhelming and often disre-garded. Some prescribers described how multiple alerts could lead to a disregard of all alerts therefore those considered important were lost amongst those perceived to be more trivial. This concords with Scott (p.569) who in a systematic review of CDS systems concluded that CDS *'need to generate advice and alerts for the right patient at the right time with the right*

*information'* [11]. Previous research has focused upon overrides of such alerts and made suggestions that the tiering of alerts, with the more severe highlighted above the less crucial could reduce the number of overrides [15].

Previous research has thus focused upon the design of alert systems and their functional nature. This problematises the design elements of the technology and sees the technology and the users as separate variables. Changes in design of technology are assumed to lead to changes in acceptance [11]. What this approach fails to capture is the reciprocal and recursive nature of technological implementation and adoption within healthcare and how implementation is a social practice. Importantly our findings highlight a more recursive understanding of technological interventions in which the technology is intertwined and interdependent with the people who use the technology and the contexts within which it is used [7, 32]. Whilst previous research has found that IT systems can undermine GP expertise and autonomy and that clinical work can be reduced to a technical activity [7, 33–35] in our study participants reported that the information the CDS provided was balanced out against the prescribers experience and expertise, with prescribers relating how they would consider alerts within broader considerations of what they knew of the patient and their medical history. This could be particularly the case for patients on longer term medications who had not had problems in the past and therefore prescribers weighed the lack of previous problems against the advice in the alerts from the system.

## Strengths and limitations

A particular strength of this study is the use of NPT, which enabled a nuanced understanding of the ways in which the CDS was implemented and embedded. NPT provided a framework that allowed us to understand the relationships involved in the implementation of the system both in the relationship of users to the technology and in the engagement and communication between different stakeholders. This allowed for a clear understanding of how the intervention involved work at different levels to maintain the system and allow for its continued sustainable use and gradual adoption into everyday practice. A further strength was the longitudinal design with a variety of stakeholders from different professions interviewed over time across multiple geographical locations. Whilst a number of different stakeholders, including prescribers in different roles (GPs, nurses and pharmacist) were interviewed, there could have been greater breadth across general practices. Many practice staff were drawn from two of the four targeted CCG areas due to recruitment difficulties related to GP time and workload. Whilst there was divergence of viewpoints across the data set with positive and negative comments there may have been some bias in the sampling. It is possible that the participants who agreed to be interviewed, in comparison to those who declined, were more active in using the system, more focused upon medication safety and therefore gave more positive responses. This bias may also have been present at CCG level where the staff who were interviewed had invested considerable time and effort in the implementation.

## Implications for future research policy and practice

Our findings that prescribers balanced out the information from the alerts against the needs of patients adds new insights to support future design, implementation and sustainability of CDS systems by recognising and providing additional information that prescribers may need, such as alert/override and prescription histories, and more intuitive alert triage and presentation approaches for complex, polypharmacy patients.

Our findings indicated a difference of focus between CCG staff and GPs around the importance of cost and safety. Whilst CCG staff discussed wanting a strategy that included patient

safety their main focus was around cost savings. For GPs it was more important that system effectively gave patient safety alerts. Future utilisation of the CDS and other similar interventions might look to mitigate against this difference of focus principally by examining the cost savings attached to safety improvement through for instance patients not requiring hospitalisation.

Our research highlighted that for this CDS system a great deal of work was undertaken at CCG level to ensure that prescribers at in GP practices received pertinent and relevant alerts and that this then reduced the possibility for alert fatigue. The careful monitoring of suites of alertswas therefore important for providing a system that would be accepted and could be maintained and sustained. CCG medicine management teams in the future might consider the importance of this profile management in sustaining this and similar interventions. This was a novel finding in that the use of CDS systems is not simply confined to the acceptance of alerts in general practices. Our research also highlighted therefore that different stakeholders undertook different activities in the implementation and use of the system with CCG staff involved in managing the CDS profile of alerts and GPs responding to the alerts as presented to them. This has important implications for future interventions and research in understanding the different work undertaken by different individuals and groups of stakeholders. Different prescribers utilised the CDS in general practices. Whilst in this qualitative work we did not explore variations in utilisation rates by different prescribers future research could further investigate the extent of use and clinical impact of this CDS and other systems. CDS implementation requires engagement from all stakeholders. Our findings have shown that a flow of communication between the CDS developers, CCGs and those working in general practice was crucial to the implementation and ongoing sustainability of the system. There were some experiences of implementation as 'top-down', and the avoidance of such approaches may support 'buy-in' from all stakeholders and increase the sustained use of the system. This could be achieved through local champions of the intervention. As pharmacist roles in general practice increase as a result of recent policy interventions [36–38] they may be well suited to enter this role. The clear positives around the ability to feedback as a two-way process at all levels was also valued and should be considered in future implementation.

Our research was informed by NPT as a theoretical framework to aid the evaluation of the complex, interdependent social processes involved in the evaluation. We feel that future research might benefit from exploring technological interventions for medication safety from such theoretical backgrounds. This will enable research to focus upon the social processes of interventions that are not necessarily linear and may help to better understand the work that people undertake to ensure that interventions are sustained in the longer term.

## Conclusions

The CDS system was understood as having a positive role in medication safety in providing relevant patient specific information to prescribers in a timely manner. Engagement between stakeholders was considered important for the initial and sustained use of the intervention by prescribers. At a strategic level, such sustained implementation might be enhanced by careful profile management of the suite of alerts in the system alongside more novel approaches to support what prescribers described as a balance between the prescribing alerts, their own judgement and the holistic needs of the patient. This was important because it indicates that the use of technology is not merely about the end users. There also needs to be more shared understanding of the purpose of the CDS between the CCGs and general practices particularly concerning safety benefits and cost saving. Importantly, we have highlighted how the use of the system operated within the contextual background of primary care. The findings here may

support further development of CDS systems particularly in the how they might have further utility and adaptability for different contexts.

## Supporting information

**S1 Appendix. Interview topic guide.**
(DOCX)

**S2 Appendix. Coding framework.**
(DOCX)

**S1 Data. Data set—All extracts from coding.**
(PDF)

**S1 Checklist. COREQ research checklist.**
(DOCX)

## Acknowledgments

We are grateful for the support for the project from the CCGs and to all interview participants who kindly gave their time. We would like to thank Professor Carl May for his expert advice and support.

## Author Contributions

**Conceptualization:** Mark Jeffries, Nde-Eshimuni Salema, Libby Laing, Azwa Shamsuddin, Aziz Sheikh, Anthony Avery, Antony Chuter, Justin Waring, Richard N. Keers.

**Data curation:** Mark Jeffries.

**Formal analysis:** Mark Jeffries, Nde-Eshimuni Salema, Libby Laing, Antony Chuter, Richard N. Keers.

**Investigation:** Mark Jeffries, Richard N. Keers.

**Methodology:** Mark Jeffries, Nde-Eshimuni Salema, Libby Laing, Azwa Shamsuddin, Aziz Sheikh, Anthony Avery, Antony Chuter, Justin Waring, Richard N. Keers.

**Project administration:** Mark Jeffries.

**Supervision:** Anthony Avery, Richard N. Keers.

**Validation:** Mark Jeffries, Nde-Eshimuni Salema, Libby Laing, Azwa Shamsuddin, Aziz Sheikh, Anthony Avery, Antony Chuter, Justin Waring, Richard N. Keers.

**Writing – original draft:** Mark Jeffries.

**Writing – review & editing:** Mark Jeffries, Nde-Eshimuni Salema, Libby Laing, Azwa Shamsuddin, Aziz Sheikh, Anthony Avery, Antony Chuter, Justin Waring, Richard N. Keers.

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
