## [Decision Letter · Decision Letter 0]

15 Dec 2020

PONE-D-20-35487

The Implementation, Use and Sustainability of a Clinical Decision Support System for Medication Optimisation in Primary Care: A Qualitative Evaluation Using Normalisation Process Theory

PLOS ONE

Dear Dr. Jeffries,

Thank you for submitting your manuscript to PLOS ONE. After careful consideration, we feel that it has merit but does not fully meet PLOS ONE’s publication criteria as it currently stands. Therefore, we invite you to submit a revised version of the manuscript that addresses the points raised during the review process.

Reviewers agreed that this work is interesting, nevertheless, more explanations are needed regarding methodology and the replicability of the results e.g., items of the interviews, coding etc. Moreover, a more detail analysis is asked by both the reviewers regarding the four themes/constructs. I suggest authors should try to comply with all the indications of both the reviewers as these are quite reasonable and well aligned. 

We look forward to receiving your revised manuscript.

Kind regards,

Simone Borsci, Ph.D.

Academic Editor

PLOS ONE

Journal Requirements:

2. When reporting the results of qualitative research, we suggest consulting the COREQ guidelines: http://intqhc.oxfordjournals.org/content/19/6/349. In this case, please consider including more information on the number of interviewers, their training and characteristics.Moreover, please provide the interview guide used as a Supplementary file.

4. Please include a copy of Table 2 which you refer to in your text on page 8.

Reviewers' comments:

Reviewer's Responses to Questions

**Comments to the Author**

1. Is the manuscript technically sound, and do the data support the conclusions?

Reviewer #1: Yes

Reviewer #2: Yes

2. Has the statistical analysis been performed appropriately and rigorously? 

Reviewer #1: N/A

Reviewer #2: N/A

3. Have the authors made all data underlying the findings in their manuscript fully available?

Reviewer #1: Yes

Reviewer #2: No

4. Is the manuscript presented in an intelligible fashion and written in standard English?

Reviewer #1: Yes

Reviewer #2: Yes

5. Review Comments to the Author

Reviewer #1: Thanks for providing the opportunity to review manuscript entitled “The Implementation, Use and Sustainability of a Clinical Decision Support System for Medication Optimisation in Primary Care: A Qualitative Evaluation Using Normalisation Process Theory”. This qualitative study evaluates the application of a clinical decision support system in general practices in the English National Health Service (NHS). I have a number of comments on the manuscript as follows.

Major comments:

1. Use of the Normalisation Process Theory: The authors have repeatedly highlighted the use of the NPT including in the title, abstract and all sections of the manuscript. However, they have neither given details of the NPT nor a strong justification of it as a theoretical framework for use in the current study. Table 1 provides includes various constructs of the NPT but there is limited explanation/description of the constructs. No source is given for Table 1.

2. Data are collected from different types of participants representing diverse organisations. A flow diagram showing the types and number of organisations and participants would be useful.

3. Data were collected by semi-structured interviews but and the interview guide/schedule was guided by the NPT but there is no information about the content of the interview guide/schedule its piloting if any.

4. The authors created codes and themes from qualitative interviews data. Inclusion of a map / dendrogram of codes and themes would help readers and audience especially senior policy makers, managers and clinicians who have very little time to read elaborate qualitative quotes reported in the manuscripts

5. Results section reports a number of themes and lengthy quotes, which have been reported under four major constructs of the NPT. Presentation of these themes in a grid comprising the NPT constructs could be helpful. NPT constructs reported in Table include sub-constructs, which might be useful to report the themes developed under the relevant sub-constructs.

6. Results show that cost saving was the main aim of the CCGs while patient safety was the main objective of GPs to use the CDS system. These are two divergent objectives, which need to be discussed, especially how different views were / could be reconciled and whether there would be any implications for addressing these issues.

7. Neither the introduction section nor the discussion section provided a review of current literature on the use, advantages and limitations of the NPT in the context of CDS system.

Minor comments

8. In the abstract, methods section does not report the number of different participating organisations, which need to be reported for each type of organisation involved. This section should report the mean/median duration of a typical interview. It is not clear what does the term ‘Staff stakeholders’ refers to. The author report that they created “themes developed into frameworks’. Should the framework be constructs?

9. BACKGROUND: Please spell out the ‘NHS’ (line 78, page 4).

10. METHODS: The authors report “Practices and CCG areas” (lines 128, page 6). Should this be ‘General practices and CCG area’?

11. METHODS: The authors report that “Semi-structured interviews were conducted with a range of stakeholders” (lines 154-55, page 6). Please specify who and how many were they.

12. Could you please add the date of ethics approval?

13. Results: There were four group interviews (lines 199, page 8). Could you please report the number of interviewees in each of these group interviews and who the participants were?

14. RESULTS: The authors mapped the main findings into 4 constructs of the NPT (liens 202-, page 8), which suggests that all themes fitted in the four constructs. Wondering whether there were there any issues that did not fit in the give NPR constructs. If so, please report them.

15. Some quotes are very long e.g. quote from CCG pharmacist 4 – lines 406-419 and GP” – lines 527-538. Long quotes could be reduced by truncating them.

16. RESULTS: The authors report “Learning was helped by depersonalised feedback.” (Lines 596, page 21). Who and how depersonalised feedback was given?

17. DICUSSION: It is hard to find the novelty and contributions of the study to the body of knowledge.

18. STRENGTHS & LIMITATIONS: The strengths are mostly around the use of NPT whereas the strengths should be about the findings of the study.

Reviewer #2: This paper presents a qualitative inquiry into factors surrounding the use of a clinical decision support system for medication. The authors described a broad effort to interview a range of participants with differing perspectives, all taken from the development, implementation, and use of a clinical decision support system in English care contexts. Interviews conducted from the perspective of Normalisation Process Theory are presented, with numerous quotes used to paint a rich picture of respondents' attitudes toward these systems. The paper is well-written, the methods are clear, and the results seem plausible. That said, I have some concerns with the methodology and the presentation. of the results.

Regarding methodology, some additional context on normalization process theory is needed. The phrase "middle-range theory" may be familiar to sociologists, but clinical and biomedical informatics readers might benefit from an introduction. Specifically, how does NPT add to theories such as UTAUT, which assesses intention to use through both individual and organizational perspectives, or distributed cognition theories discussing how systems are used in organizational contexts? None of this is to say that NPT is inappropriate, but the paper would benefit from some deeper discussion of the methods.

The description of the results made for an interesting read - the quotes were informative and thought-provoking. However, the results section read more like a preliminary draft than a finished paper - far too many quotes and far too little structure. Although I can certainly understand the desire to include too much discussion in the results section, I would have found it helpful to have more summary and analysis. Perhaps some tables summarizing key themes under each of the four constructs? Also, there was very little clarity of how these concepts and observations differed across the different groups of respondents.

These difficulties persist into the discussion. Although the authors appropriately provide a nice description of how their results are consistent with observations from prior studies, there is less clarity as to the novel insights provided by this paper. It would be helpful if the authors were able to summarize their contributions more concisely.

Regarding the limitations of this study, I am concerned that the authors do not include any discussion of the utilization of the systems involved. Extensive prior literature, and some of the comments in this paper, clearly show that clinicians often disregard drug related alerts. This suggests that there may be a disconnect - it's entirely possible that some of the positive statements about the CDS systems might have, in fact, been made by clinicians who regularly dismissed or ignored system recommendations. Ideally, this qualitative detail might be accompanied by some assessments of the rates of utilization of the system, but I understand that this data might be hard to get. However, the authors should consider a careful look into the possibility that the interview results are biased, particularly since they were conducted outside of care contexts and may be impacted by demand characteristics (ie, the tendency of respondents to answer in a manner that might make them "look good" to the interviewers). Some discussion of these questions would be appropriate.

Finally, regarding data sharing, more detail is needed. Even if transcripts of interviews cannot be shared, the authors should be able to share their codebook, any summary statistics, and other related material.

- data sharing

---

## [Author Response · Author response to Decision Letter 0]

14 Mar 2021

PONE-D-20-35487

The Implementation, Use and Sustainability of a Clinical Decision Support System for Medication Optimisation in Primary Care: A Qualitative Evaluation Using Normalisation Process Theory

PLOS ONE

Dear Dr Borsci, 

Re: ‘The Implementation, Use and Sustainability of a Clinical Decision Support System for Medication Optimisation in Primary Care: A Qualitative Evaluation Using Normalisation Process Theory’

Thank you for the thoughtful and constructive feedback on our manuscript, which we have carefully considered. We have in the light of this feedback made a number of revisions to the manuscript, which are detailed in the table below.

We are confident that these revisions will be to your satisfaction, but please do contact me if you require any further clarification or revisions. 

Yours sincerely,

Mark Jeffries, on behalf of the co-authors

 

Editor comments Response/Changes made Location of change in tracked changed manuscript

https://journals.plos.org/plosone/s/file?id=ba62/PLOSOne_formatting_sample_title_authors_affiliations.pdf We have checked the formatting n/a

2. When reporting the results of qualitative research, we suggest consulting the COREQ guidelines: http://intqhc.oxfordjournals.org/content/19/6/349. In this case, please consider including more information on the number of interviewers, their training and characteristics. Moreover, please provide the interview guide used as a Supplementary file. We have amended the COREQ and added the detail about the interviewer to the manuscript. In the study design section of the methods, we have added the following: 

“All interviews were conducted solely by MJ, a researcher with extensive experience in conducting research related to medication safety and the use of technology and expertise in qualitative research interviewing. Interviews took place at the participants’ usual place of work (general practice, CCG offices) or at university premises (two interviews). No interviewees were known to MJ prior to the interview.” 

The interview guide is now added as a supplementary file. P7 Lines 166 -171

We will update your Data Availability statement to reflect the information you provide in your cover letter. We have included two supplementary files. One detailing the coding framework and one the codebook showing how those codes matched to quotations from participants.

We have changed our data availability statement which now reads.

“We have provided a supplementary file of the coding framework. In addition we have provided a supplementary file of all the extracts from the transcripts detailed per theme, sub theme and code. This extensive and detailed document provides a data set from which the study can be fully replicated. 

Full transcripts are not available to preserve the anonymity of participants as per our ethics. This is a qualitative study confined to relatively small groups of health care professionals in specific roles, particularly CCG staff. Making the full data set publicly available could therefore potentially lead to the identification of participants. Our ethics approval was granted based on the anonymity of the individuals consenting to participate and specifically referred to only anonymised quotations from transcripts being made available as we have in the supplementary file. As such the participants did not consent to full their transcript being made publicly available"

 Cover letter

4. Please include a copy of Table 2 which you refer to in your text on page 8. This was originally a typo and should have read Table 1, however we have now included a further tables. Table 1 details the NPT constructs, Table 2 the participants and Table 3 how the findings here map to the NPT constructs. Page 8

 We have added captions for our supporting files at the end of the manuscript. Page 36

Reviewer One 

Major Comments 

1. Use of the Normalisation Process Theory: The authors have repeatedly highlighted the use of the NPT including in the title, abstract and all sections of the manuscript. However, they have neither given details of the NPT nor a strong justification of it as a theoretical framework for use in the current study. Table 1 provides includes various constructs of the NPT but there is limited explanation/description of the constructs. No source is given for Table 1. Thank you for these helpful comments. We have included additional text to help the reader with further details of NPT. We have added to the justification for why we have used it here. The subsection in the Methods on page 7 detailing the theoretical framework now reads: 

“Theoretical frameworks for technological interventions have seen technology, people and organisations as separate things which operate at multiple levels but do not do so through interaction and have often regarded these as separate homogenous elements. [7]. Technological intervention can be observed in ways that understand it as part of complex and interdependent social processes. [6,7] Normalization Process Theory (NPT) has been used to understand the ways new technologies and work practices are implemented and focuses upon the social processes of implementation. [4, 22] Interventions involve changes in the actions that people do so that the new behaviours involved in the new intervention become adopted. [20] NPT suggests that interventions are adopted and implemented because of the work and actions people undertake, both individually and collectively, to integrate the intervention into routine practice. [20, 23-25] NPT provides a rational framework and that allows for an understanding of how individuals and groups make sense of a new intervention and work together to build confidence in the new practice and change their actions and behaviours in order to enable it to happen. [20, 23-25] Implementation of a new intervention involves changes in human interactions with material things, changes in human relationships, changes in the rules and social norms that make actions possible and changes in the ways systems are understood and defined. [25] 

NPT is built upon four constructs: coherence – the ways in which the intervention is understood, cognitive participation – the ways people work together to put the new intervention into practice, collective action – the ways people work to operationalise the new intervention, and reflexive monitoring – how people evaluate the new practice and work to sustain it. These constructs are further as described in Table 1. NPT has been used in a number of evaluations of health care interventions including those for prescribing safety and patient digital feedback and has been said to be particularly useful for drawing out the multi-faceted nature of interventions. [4, 26, 27] In recent research NPT was considered useful in providing insights into how relationships and communication between health professionals could help drive the implementation and sustainability of a technological intervention utilising an audit and feedback system for medication safety in primary care. [4] It was therefore considered an appropriate theoretical framework through which to explore the use of a CDS in primary care”

We have now provided the sources for Table 1 Pages 7, line 175 -Page 8, line 205

Page 35

2. Data are collected from different types of participants representing diverse organisations. A flow diagram showing the types and number of organisations and participants would be useful.

 We have provided a table (Table 2) of participants broken down by role and CCG region. Page 36

3. Data were collected by semi-structured interviews but and the interview guide/schedule was guided by the NPT but there is no information about the content of the interview guide/schedule its piloting if any.

 Thank you for this helpful comment we agree that more was needed here. The interview guide was informed by NPT and was developed from previous literature. The authors have extensive previous experience of developing topic guides for similar studies (eg Reference 4 Jeffries et al PLOS ONE 2018) and drew upon this experience. The interview was not piloted but was reviewed throughout data collection. We have changed the wording of the first part of the data collection section to reflect this. This now reads:

“The semi-structured interview schedule was informed by Normalisation Process Theory (NPT) [22] and developed through and discussions between the authors and reading of relevant literature. [4-8, 14]. The interview schedule (included as supplementary file S1 Appendix – Interview Topic Guide) was designed to illicit how people understood the value of the CDS, how it had been implemented, the work in using the system and how the intervention might be sustained. The interview schedule was reviewed iteratively throughout data collection to ensure it continued to be appropriate”.

We have also included the interview schedule as a supplementary file. 

Page 6, line 155 - Page 7 line 161

Page 37

4. The authors created codes and themes from qualitative interviews data. Inclusion of a map / dendrogram of codes and themes would help readers and audience especially senior policy makers, managers and clinicians who have very little time to read elaborate qualitative quotes reported in the manuscripts We have included two supplementary files. One detailing the coding framework and one the codebook showing how those codes matched to quotations from participants. Page 37

5. Results section reports a number of themes and lengthy quotes, which have been reported under four major constructs of the NPT. Presentation of these themes in a grid comprising the NPT constructs could be helpful. NPT constructs reported in Table include sub-constructs, which might be useful to report the themes developed under the relevant sub-constructs. We agree that this would be helpful. We have included a new table (Table 3) which maps our findings to the NPT constructs. The findings mapped broadly onto the constructs but not onto each component or subconstruct as detailed in the table.

In addition, in the results we have added a description after the subheading for each NPT construct as to how the themes were mapped to the components in that construct. Page 36

Page 9 Lines 246-252; Page 16 Lines 452-454; Page 19 Lines 535-538; Page 23 lines 646-648.

6. Results show that cost saving was the main aim of the CCGs while patient safety was the main objective of GPs to use the CDS system. These are two divergent objectives, which need to be discussed, especially how different views were / could be reconciled and whether there would be any implications for addressing these issues. Thank you for this helpful point. We have reflected in the discussion. We have created a new subsection here focusing upon the implications of our study on future research policy and practice. To this we have added the following paragraph:

“Our findings indicated a difference of focus between CCG staff and GPs around the importance of cost and safety. Whilst CCG staff discussed wanting a strategy that included patient safety their main focus was around cost savings. For GPs it was more important that system effectively gave patient safety alerts. Future utilisation of the CDS and other similar interventions might look to mitigate against this difference of focus principally by examining the cost savings attached to safety improvement through for instance patients not requiring hospitalisation.”

In addition, we have added this sentence to the conclusions:

“There also needs to be more shared understanding of the purpose of the CDS between the CCGs and general practices particularly concerning safety benefits and cost saving”.

 Page 28-29

Page 28 Lines 803-809

Page 30 861-864

7. Neither the introduction section nor the discussion section provided a review of current literature on the use, advantages and limitations of the NPT in the context of CDS system. We have extensively rewritten the section on NPT in the methods (please see our response to point 1 above) and this now has references to other literature that have used NPT and a justification for the use here. We also now reference NPT in the introduction. 

The merits of using NPT are already considered in the discussion but we have added to this. To the strengths and limitations subsection we have added:

 “NPT provided a framework that allowed us to understand the relationships involved in the implementation of the system both in the relationship of users to the technology and in the engagement and communication between different stakeholders.” 

We have also added to the new subsection on implications a paragraph about the use of such theoretical frameworks. This reads:

“Our research was informed by NPT as a theoretical framework to aid the evaluation of the complex, interdependent social processes involved in the evaluation. We feel that future research might benefit from exploring technological interventions for medication safety from such theoretical backgrounds. This will enable research to focus upon social processes that are not necessarily linear and may help to better understand the work that people undertake to ensure that interventions are sustained in the longer term.”

 Introduction Page 5 lines 106 -108 Methods Pages 7, line 175 - Page 8, line 205.

Page 27 Lines 780-782

Page 29 Lines 835-840

Minor comments

9. BACKGROUND: Please spell out the ‘NHS’ (line 78, page 4). Thank you. We have made this change Page 4 line 78

10. METHODS: The authors report “Practices and CCG areas” (lines 128, page 6). Should this be ‘General practices and CCG area’? We have changed this as you suggest. Thank you. Page 6 line 131

11. METHODS: The authors report that “Semi-structured interviews were conducted with a range of stakeholders” (lines 154-55, page 6). Please specify who and how many were they. The different stakeholders group and participant numbers are detailed at the start of the results and now in table 2 that we have included in response to your comment 2 above. We do however agree it would be useful to have a general comment on who these stakeholders were here and have revised the text accordingly. This now reads: 

“Semi-structured interviews were conducted with a range of stakeholders – CCG staff including Pharmacists and Pharmacy Technicians, GP Staff including GPs, Nurses and Pharmacists and staff from the software developers.” Results Page 9, 235-240

Methods Page 7, Lines 161-163

12. Could you please add the date of ethics approval? We have added the dates of our ethical and HRA approval. Methods Page 9 Lines 222-224

13. Results: There were four group interviews (lines 199, page 8). Could you please report the number of interviewees in each of these group interviews and who the participants were? There were in fact 5 group interviews. We have added the following text to the start of the results.

“The 5 group interviews each involved staff working at the same CCG or general practice (Group interview 1 - CCG pharmacist and CCG Pharmacy Technician; Group interview 2 – 3 GPs; Group interview 3 - GP and GP Pharmacist Group interview 4 CCG Pharmacist and CCG data analyst Group interview 5 Follow up to group interview 3 with GP and GP Pharmacist. Stakeholder interviews are further described in table 2.” Results Page 9, 235-240

14. RESULTS: The authors mapped the main findings into 4 constructs of the NPT (liens 202-, page 8), which suggests that all themes fitted in the four constructs. Wondering whether there were there any issues that did not fit in the give NPR constructs. If so, please report them. Thank you for this point. We did not find themes that we were not able to map onto the NPT constructs. We trust that table 3 helps to clarify how the results mapped to the NPT constructs. 

15. Some quotes are very long e.g. quote from CCG pharmacist 4 – lines 406-419 and GP” – lines 527-538. Long quotes could be reduced by truncating them. We agree that the manuscript would be improved by reducing the length of some of the quotations. We have made several changes to the results, truncating or deleting quotations and rewriting several sections as detailed in the revised manuscript. Results -Pages 9 -25

16. RESULTS: The authors report “Learning was helped by depersonalised feedback.” (Lines 596, page 21). Who and how depersonalised feedback was given? We have deleted the comment about depersonalised feedback as on reflection it does not fit with our comments in the results at that point or the quotation from the prescriber. Thank you for this helpful comment. Page 24 lines 688

17. DICUSSION: It is hard to find the novelty and contributions of the study to the body of knowledge. We have rewritten several parts of the discussion to provide some clarity.

We have restructured the discussion and included a subsection detailing the implications for future research policy and practice. We feel this section now draws the reader to the important and novel findings of our study and how it raises implications. 

This subsection reads:- 

“Our findings that prescribers balanced out the information from the alerts against the needs of patients adds new insights to support future design, implementation and sustainability of CDS systems by recognising and providing additional information that prescribers may need, such as alert/override and prescription histories, and more intuitive alert triage and presentation approaches for complex, polypharmacy patients. 

Our findings indicated a difference of focus between CCG staff and GPs around the importance of cost and safety. Whilst CCG staff discussed wanting a strategy that included patient safety their main focus was around cost savings. For GPs it was more important that system effectively gave patient safety alerts. Future utilisation of the CDS and other similar interventions might look to mitigate against this difference of focus principally by examining the cost savings attached to safety improvement through for instance patients not requiring hospitalisation.

Our research highlighted that for this CDS system a great deal of work was undertaken at CCG level to ensure that prescribers at in GP practices received pertinent and relevant alerts and that this then reduced the possibility for alert fatigue. The careful monitoring of suites of alerts was therefore important for providing a system that would be accepted and could be maintained and sustained. CCG medicine management teams in the future might consider the importance of this profile management in sustaining this and similar interventions. Our research also highlighted therefore that different stakeholders undertook different activities in the implementation and use of the system with CCG staff involved in managing the CDS profile of alerts and GPs responding to the alerts as presented to them. This has important implications for future interventions and research in understanding the different work undertaken by different individuals and groups of stakeholders. Different prescribers utilised the CDS in general practices. Whilst in this qualitative work we did not explore variations in utilisation rates by different prescribers future research could further investigate the extent of use and clinical impact of this CDS and other systems. 

CDS implementation requires engagement from all stakeholders. Our findings have shown that a flow of communication between the software developers, CCGs and those working in general practice was crucial to the implementation and ongoing sustainability of the system. There were some experiences of implementation as ‘top-down’, and the avoidance of such approaches may support ‘buy-in’ from all stakeholders and increase the sustained use of the system. This could be achieved through local champions of the intervention. As pharmacist roles in general practice increase as a result of recent policy interventions [36, 37, 38, ] they may be well suited to enter this role. The clear positives around the ability to feedback as a two-way process at all levels was also valued and should be considered in future implementation.

Our research was informed by NPT as a theoretical framework to aid the evaluation of the complex, interdependent social processes involved in the evaluation. We feel that future research might benefit from exploring technological interventions for medication safety from such theoretical backgrounds. This will enable research to focus upon the social processes of interventions that are not necessarily linear and may help to better understand the work that people undertake to ensure that interventions are sustained in the longer term.”

We have also added to the conclusion, which now reads:

“The CDS system was understood as having a positive role in medication safety in providing relevant patient specific information to prescribers in a timely manner. Engagement between stakeholders was considered important for the initial and sustained use of the intervention by prescribers. At a strategic level, such sustained implementation might be enhanced by careful profile management of the suite of alerts in the system alongside more novel approaches to support what prescribers described as a balance between the prescribing alerts, their own judgement and the holistic needs of the patient. This was important because it indicates that the use of technology is not merely about the end users. There also needs to be more shared understanding of the purpose of the CDS between the CCGs and general practices particularly concerning safety benefits and cost saving. Importantly, we have highlighted how the use of the system operated within the contextual background of primary care. The findings here may support further development of CDS systems particularly in the how they might have further utility and adaptability for different contexts.” Page 25 lines 719-732, Page 26 Lines 756-758, Page 27 Lines 763-765

Pages 28 Line 797 -Page 29 line 840

Page 29 Line 855 - Page 30 Line 867

18. STRENGTHS & LIMITATIONS: The strengths are mostly around the use of NPT whereas the strengths should be about the findings of the study. We think that it is standard practice for the ‘strengths and limitations’ section of the discussion to focus on the research methodology (and methods), which in our case includes our theoretical framework. From our experience, authors do not usually focus on the findings in this section of a manuscript. Therefore, we do not plan to include the findings of the study in this section. 

Reviewer 2 

 Regarding methodology, some additional context on normalization process theory is needed. The phrase "middle-range theory" may be familiar to sociologists, but clinical and biomedical informatics readers might benefit from an introduction. Specifically, how does NPT add to theories such as UTAUT, which assesses intention to use through both individual and organizational perspectives, or distributed cognition theories discussing how systems are used in organizational contexts? None of this is to say that NPT is inappropriate, but the paper would benefit from some deeper discussion of the methods.

 Thank you for the helpful comments around our theoretical approach. We felt that NPT would be most appropriate because it is a framework that helps to understand the collective work that people do. It also is not limited to technology and whilst our paper is about a technological intervention the focus here is on implementation, sustainability and the embedding of it into everyday use. We feel that NPT captured the important communication and collaborative aspects of the intervention in relation to this focus. We also feel that distributed cognition theories would be more about individual responses to using the technology, whilst accepting that would be within an organisational context. UTAUT like other acceptance theories would see technology and people as discrete entities. In contrast with NPT we felt we could focus more holistically on the social processes involved in the intervention and more upon the interaction and interdependence of technology and users. We have changed the section on NPT in the methods and included more detail. We trust the changes we have made make this clearer. Please see our response to reviewer 1 point above. 

 Pages 7, line 175 -Page 8, line 205

The description of the results made for an interesting read - the quotes were informative and thought-provoking. However, the results section read more like a preliminary draft than a finished paper - far too many quotes and far too little structure. Although I can certainly understand the desire to include too much discussion in the results section, I would have found it helpful to have more summary and analysis. Perhaps some tables summarizing key themes under each of the four constructs? Also, there was very little clarity of how these concepts and observations differed across the different groups of respondents.

 We have included more summary analysis of the results within the results section to provide further structure. In addition, in the results we have added a description after the subheading for each NPT construct as to how the themes were mapped to the components in that construct. The quotations have been shortened or deleted where possible. We feel this now provides more structure and greater clarity to the results. We have also included a table (table 3) summarizing the themes within the NPT constructs. We have now reflected upon differences between stakeholders and stakeholder groups in the discussion particularly in the new subsection on implications for future research policy and practice. Please see response to reviewer 1 point 17 above. Results - Page 9 Lines 246-252; Page 16 Lines 452-454; Page 19 Lines 535-538; Page 23 lines 646-648.

Implications for future research policy and practice Pages 28 Line 797 -Page 29 line 840

These difficulties persist into the discussion. Although the authors appropriately provide a nice description of how their results are consistent with observations from prior studies, there is less clarity as to the novel insights provided by this paper. It would be helpful if the authors were able to summarize their contributions more concisely. We have made changes in the discussion to address this. We have rewritten parts of the first and third paragraphs of the discussion to provide clarity on the key findings of our research and provide more clarity. 

We have created a new subsection outlining the implications for future research policy and practice. Please see response to reviewer 1 point 17 above. Discussion Page 25 lines 719-732, Page 26 Lines 756-758, Page 27 Lines 763-765.

Implications for future research policy and practice Pages 28 Line 797 -Page 29 line 840

Regarding the limitations of this study, I am concerned that the authors do not include any discussion of the utilization of the systems involved. Extensive prior literature, and some of the comments in this paper, clearly show that clinicians often disregard drug related alerts. This suggests that there may be a disconnect - it's entirely possible that some of the positive statements about the CDS systems might have, in fact, been made by clinicians who regularly dismissed or ignored system recommendations. Ideally, this qualitative detail might be accompanied by some assessments of the rates of utilization of the system, but I understand that this data might be hard to get. However, the authors should consider a careful look into the possibility that the interview results are biased, particularly since they were conducted outside of care contexts and may be impacted by demand characteristics (ie, the tendency of respondents to answer in a manner that might make them "look good" to the interviewers). Some discussion of these questions would be appropriate. Thank you for these helpful points. We feel that prescribers gave both negative and positive responses in interviews and there was a variation of views across the data set. To reflect upon the possibility of bias we have added the following to the strengths and limitations section:

“Whilst there was divergence of viewpoints across the data set with positive and negative comments there may have been some bias in the sampling. It is possible that the participants who agreed to be interviewed, in comparison to those who declined, were more active in using the system, more focused upon medication safety and therefore gave more positive responses. This bias may also have been present at CCG level where the staff who were interviewed had invested considerable time and effort in the implementation”. 

In the new subsection on implications for future research, policy and practice where we discuss differences and a disconnect between different stakeholders with regard to cost and safety, and how different stakeholders were involved in the intervention in different ways – CCG staff maintain the profile of alerts and GP staff responding to alerts in practice.. Here we make recommendations for future implementation. Please see response to reviewer 1 point 17 above

 Whilst in this qualitative work we did not explore variations in utilisation rates by different prescribers we agree that future research could further investigate the extent of use and clinical impact of this CDS and other systems. We have added to the implications subsection to reflect on this. 

Page 2, lines 787-793

Implications for future research policy and practice Pages 28 Line 797 -Page 29 line 840

Finally, regarding data sharing, more detail is needed. Even if transcripts of interviews cannot be shared, the authors should be able to share their codebook, any summary statistics, and other related material. We have included two supplementary files. One detailing the coding framework and one the codebook showing how those codes matched to quotations from participants.

---

## [Decision Letter · Decision Letter 1]

11 Apr 2021

PONE-D-20-35487R1

The Implementation, Use and Sustainability of a Clinical Decision Support System for Medication Optimisation in Primary Care: A Qualitative Evaluation Using Normalisation Process Theory

PLOS ONE

Dear Dr. Jeffries,

Thank you for submitting your manuscript to PLOS ONE. After careful consideration, we feel that it has merit but does not fully meet PLOS ONE’s publication criteria as it currently stands. Therefore, we invite you to submit a revised version of the manuscript that addresses the points raised during the review process.

One of the reviewers still has some comments which I believe are helpful to further improve the manuscript. Please revise in tune, and resubmit. Then I will personally check the revision without involving the reviewer and if satisfactory I will accept the article.

We look forward to receiving your revised manuscript.

Kind regards,

Simone Borsci, Ph.D.

Academic Editor

PLOS ONE

Journal Requirements:

Reviewers' comments:

Reviewer's Responses to Questions

**Comments to the Author**

1. If the authors have adequately addressed your comments raised in a previous round of review and you feel that this manuscript is now acceptable for publication, you may indicate that here to bypass the “Comments to the Author” section, enter your conflict of interest statement in the “Confidential to Editor” section, and submit your "Accept" recommendation.

Reviewer #1: (No Response)

Reviewer #2: All comments have been addressed

2. Is the manuscript technically sound, and do the data support the conclusions?

Reviewer #1: Yes

Reviewer #2: Yes

3. Has the statistical analysis been performed appropriately and rigorously? 

Reviewer #1: N/A

Reviewer #2: N/A

4. Have the authors made all data underlying the findings in their manuscript fully available?

Reviewer #1: No

Reviewer #2: Yes

5. Is the manuscript presented in an intelligible fashion and written in standard English?

Reviewer #1: Yes

Reviewer #2: Yes

6. Review Comments to the Author

Reviewer #1: Thanks to the authors for revising and resubmitting the manuscript entitled “The Implementation, Use and Sustainability of a Clinical Decision Support System for Medication Optimisation in Primary Care: A Qualitative Evaluation Using Normalisation Process Theory”, which is improved. However, it needs further revision as suggested below.

TITLE:

The title of the manuscript is too long and it needs to be shortened such as removing ‘Using Normalisation Process Theory’.

ABSTRACT:

The abstracts need improvement especially the findings section by reporting the key findings as per the major themes and / or constructs of the Normalisation Process Theory, as reported in Table 3 and the main text. For example: COHERENCE (purpose of the system like Enhancing medication safety and improving cost effectiveness; Relationship of users to the technology; Engagement and communication between different stakeholders), COGNITIVE PARTICIPATION (Management of the profile of alerts) , COLLECTIVE ACTION (Prescribing in general practice; patient and population characteristics and engagement with patients; knowledge) REFLEXIVE MONITORING (Sustaining the use of the CDS through maintenance and customisation; Learning and behaviour change).

The authors also mention that “there was a range of contextual factors’, which have not been reported. Could the authors the key contextual factors identified.

In addition, background and methods sections in the abstract may be shortened.

Conclusion: The conclusion only focuses on the safety and alerts but it does not reflect anything about cost saving which was the primary aim of the CCGs. This aspect needs to be reflected in the conclusion in the abstract, as reported in the conclusion section in the main text.

BACKGROUND:

CDC (page 4, lines 84-85): Could the authors add a citation/reference about the CSD system evaluated. In addition, the authors call the CDS sometimes as a system (page 4, line 87 and so on) and sometimes as software (page 4, lines 84-85). Is it a system or software? Please be precise and consistent.

CCG (page 4, line 87), Please spell out the acronym CCG here as it is mentioned for the first time, which has been spelled out later on page 5 (line 123).

METHODS:

National Health Service (Pages 5-, lines 125-126): Acronym for the term ‘National Health Service’ has been given already in page 4, line 78; hence use only the acronym here (lines 125-126, page 5-6)

Follow-up interviews (page 7, lines 1163-64): Follow-up interviews were conducted about 12 months after the first interviews. This is a long period. Did the follow-up interviews were conducted with the same persons / interviewees working on the same positions as in the first interviews? Were there any new people, and how many, involved in the follow-up interviews.

Shopping vouchers (page 7, lines 172-173): The authors paid £20 shopping vouchers to participants. Was the payment included in the ethics application? Did the authors paid/ gave vouchers to all participants or some participants and who were they. It seems unbelievable that a GP will accept/ask for £20 voucher for an interview! Most of the participants were employees of either the NHS or CCGs hence paid by them so why they were reimbursed for their time in the study?

Acronym of Normalisation Process Theory: The authors have reported Normalisation Process Theory (NPT) on page 4, lines 107-08 and again they report Normalisation Process Theory (NPT) on page 7, line 179). Please report only NPT on page 7, line 179.

DISCUSSION:

Strengths and limitations (page 27, lines 776-777): The authors have stated that This was the first qualitative study to utilise NPT to understand the complex processes that influenced the implementation and sustained use in primary care of a CDS system“. However, in the Methods section - Theoretical Framework: Normalization process theory (page 8, lines 192-205), they have reported that the NPT has been used in the evaluations of several health care interventional studies so why this study is the first to use the NPT in this study. The authors may check this claim.

Reviewer #2: (No Response)

7. PLOS authors have the option to publish the peer review history of their article (what does this mean?). If published, this will include your full peer review and any attached files.

Reviewer #1: **Yes: **Syed Ghulam Sarwar Shah

Reviewer #2: **Yes: **Harry Hochheiser

---

## [Author Response · Author response to Decision Letter 1]

15 Apr 2021

PONE-D-20-35487

The Implementation, Use and Sustainability of a Clinical Decision Support System for Medication Optimisation in Primary Care: A Qualitative Evaluation Using Normalisation Process Theory

PLOS ONE

Dear Dr Borsci, 

Re: ‘The Implementation, Use and Sustainability of a Clinical Decision Support System for Medication Optimisation in Primary Care: A Qualitative Evaluation Using Normalisation Process Theory’

Thank you for the further feedback on our manuscript, which we have carefully considered. We have in the light of this feedback made further revisions to the manuscript, which are detailed in the table below.

We are confident that these revisions will be to your satisfaction, but please do contact me if you require any further clarification or revisions. 

Yours sincerely,

Mark Jeffries, on behalf of the co-authors

 

Reviewer comments Response/Changes made Location of change in tracked changed manuscript

TITLE: 

The title of the manuscript is too long and it needs to be shortened such as removing ‘Using Normalisation Process Theory’.

 We have taken appreciate that the title is quite long. We are therefore happy to change the title to: “The Implementation, Use and Sustainability of a Clinical Decision Support System for Medication Optimisation in Primary Care: A Qualitative Evaluation” Title - Page 1

ABSTRACT: 

The abstracts need improvement especially the findings section by reporting the key findings as per the major themes and / or constructs of the Normalisation Process Theory, as reported in Table 3 and the main text. For example: COHERENCE (purpose of the system like Enhancing medication safety and improving cost effectiveness; Relationship of users to the technology; Engagement and communication between different stakeholders), COGNITIVE PARTICIPATION (Management of the profile of alerts) , COLLECTIVE ACTION (Prescribing in general practice; patient and population characteristics and engagement with patients; knowledge) REFLEXIVE MONITORING (Sustaining the use of the CDS through maintenance and customisation; Learning and behaviour change).

 Thank you for this point. We have rewritten the results section of the abstract which now reads:

“Thirty-nine interviews were conducted either individually or in groups, with 33 stakeholders, including 11 follow-up interviews. Eight themes were interpreted in alignment with the four NPT constructs : Coherence (The purpose of the CDS: Enhancing medication safety and improving cost effectiveness; Relationship of users to the technology; Engagement and communication between different stakeholders); Cognitive Participation (Management of the profile of alerts); Collective Action (Prescribing in general practice, patient and population characteristics and engagement with patients; Knowledge);and Reflexive Monitoring (Sustaining the use of the CDS through maintenance and customisation; Learning and behaviour change. Participants saw that the CDS could have a role in enhancing medication safety and in the quality of care. Engagement through communication and support for local primary care providers and management leaders was considered important for successful implementation. Management of prescribing alert profiles for general practices was a dynamic process evolving over time. At regional management levels, work was required to adapt, and modify the system to optimise its use in practice and fulfil local priorities. Contextual factors, including patient and population characteristics, could impact upon the decision-making processes of prescribers influencing the response to alerts. The CDS could operate as a knowledge base allowing prescribers access to evidence-based information that they otherwise would not have.” Abstract - Page 2-3

The authors also mention that “there was a range of contextual factors’, which have not been reported. Could the authors the key contextual factors identified.

 We have added to the abstract a phrase the outlining examples of the contextual factors. Please see the revised results section of the abstract above Abstract - Page 2-3

In addition, background and methods sections in the abstract may be shortened.

 We have shortened the background and methods sections of the abstract. These now read: 

“Background

The quality and safety of prescribing in general practice is important, Clinical decision support (CDS) systems can be used which present alerts to health professionals when prescribing in order to identify patients at risk of potentially hazardous prescribing. It is known that such computerised alerts may improve the safety of prescribing in hospitals but their implementation and sustainable use in general practice is less well understood. We aimed to understand the factors that influenced the successful implementation and sustained use in primary care of a CDS system. 

Methods

Participants were purposively recruited from Clinical Commissioning Groups (CCGs) and general practices in the North West and East Midlands regions of England and from the CDS developers. We conducted face-to-face and telephone-based semi-structured qualitative interviews with staff stakeholders. A selection of participants was interviewed longitudinally to explore the further sustainability 1-2 years after implementation of the CDS system. The analysis, informed by Normalisation Process Theory (NPT), was thematic, iterative and conducted alongside data collection.” Abstract – Page 2

Conclusion: The conclusion only focuses on the safety and alerts but it does not reflect anything about cost saving which was the primary aim of the CCGs. This aspect needs to be reflected in the conclusion in the abstract, as reported in the conclusion section in the main text.

 We have adapted the conclusion of the abstract to reflect the cost-saving aspect of the CDS which we agree was important. We have added a final sentence that reads: 

“Shared understanding of the purpose of the CDS between CCGS and general practices particularly in balancing cost saving and safety messages could be beneficial” Abstract – Page 3

BACKGROUND:

CDC (page 4, lines 84-85): Could the authors add a citation/reference about the CSD system evaluated. In addition, the authors call the CDS sometimes as a system (page 4, line 87 and so on) and sometimes as software (page 4, lines 84-85). Is it a system or software? Please be precise and consistent. Regretfully we are unable to provide a reference about the CDS system. This it is a commercial product and prior agreement with the developers of the CDS was that product name should remain anonymous. We feel the findings provide recommendations for the implementations of CDS in general as well as this specific system. 

We agree that the use of system and software is inconsistent and imprecise. We have removed the use of ‘software’ throughout the manuscript. 

CCG (page 4, line 87), Please spell out the acronym CCG here as it is mentioned for the first time, which has been spelled out later on page 5 (line 123).

 Thank you. We have made this change. Page 4 

Page 5

METHODS:

National Health Service (Pages 5-, lines 125-126): Acronym for the term ‘National Health Service’ has been given already in page 4, line 78; hence use only the acronym here (lines 125-126, page 5-6)

 Thank you we have made this change. Methods - Page 5

Follow-up interviews (page 7, lines 1163-64): Follow-up interviews were conducted about 12 months after the first interviews. This is a long period. Did the follow-up interviews were conducted with the same persons / interviewees working on the same positions as in the first interviews? Were there any new people, and how many, involved in the follow-up interviews. The interval before the follow-up interviews was designed to capture use after the initial implementation of the system. The same participants took part. We have added details of the follow up interviews in the method section. This now reads:

“Follow-up interviews were conducted approximately 12 months after the first interview with 11 participants who had taken part in first interviews., These participants were purposively selected to provide a range of different stakeholders, in order to understand changes that may have occurred and how the intervention was being sustained in everyday practice beyond the initial intervention period.” Methods – page 7

Shopping vouchers (page 7, lines 172-173): The authors paid £20 shopping vouchers to participants. Was the payment included in the ethics application? Did the authors paid/ gave vouchers to all participants or some participants and who were they. It seems unbelievable that a GP will accept/ask for £20 voucher for an interview! Most of the participants were employees of either the NHS or CCGs hence paid by them so why they were reimbursed for their time in the study?

 Yes the payment was included in the ethics application and approvals. Resource consideration was a reason given for non-participation from general practices therefore the £20 voucher was useful as an incentive to take part. In our experience it is usual practice to compensate interview participants for their time. The GPs who took part in the study took time out from their working day. All CCG staff declined the vouchers. We have added this to the methods section to clarify this further.

“As approved by the ethics committee participants were offered a £20 shopping voucher per interview as reimbursement for their time. Vouchers were accepted by all staff in general practices who took time out from their working day to participate but declined by eight of the ten CCG staff.” Methods – page 7

DISCUSSION:

Strengths and limitations (page 27, lines 776-777): The authors have stated that This was the first qualitative study to utilise NPT to understand the complex processes that influenced the implementation and sustained use in primary care of a CDS system“. However, in the Methods section - Theoretical Framework: Normalization process theory (page 8, lines 192-205), they have reported that the NPT has been used in the evaluations of several health care interventional studies so why this study is the first to use the NPT in this study. The authors may check this claim.

 We agree that this sentence is inappropriate and have removed this claim. Discussion – Page 27

---

## [Editor Report · Decision Letter 2]

19 Apr 2021

The Implementation, Use and Sustainability of a Clinical Decision Support System for Medication Optimisation in Primary Care: A Qualitative Evaluation

PONE-D-20-35487R2

Dear Dr. Jeffries,

We’re pleased to inform you that your manuscript has been judged scientifically suitable for publication and will be formally accepted for publication once it meets all outstanding technical requirements.

Kind regards,

Simone Borsci, Ph.D.

Academic Editor

PLOS ONE

---

## [Editor Report · Acceptance letter]

22 Apr 2021

PONE-D-20-35487R2 

The Implementation, Use and Sustainability of a Clinical Decision Support System for Medication Optimisation in Primary Care: A Qualitative Evaluation 

Dear Dr. Jeffries:

I'm pleased to inform you that your manuscript has been deemed suitable for publication in PLOS ONE. Congratulations! Your manuscript is now with our production department. 

Kind regards, 

on behalf of

Dr. Simone Borsci 

Academic Editor

PLOS ONE